

# Two new species of *Pseudopimelodus* Bleeker, 1858 (Siluriformes: Pseudopimelodidae) from the Magdalena Basin, Colombia

Ana M. Restrepo-Gómez[1,*], José D. Rangel-Medrano[2,*], Edna J. Márquez[2] and Armando Ortega-Lara[3,4]

[1] Facultad de Ciencias Agrarias, Universidad Nacional de Colombia, Medellín, Antioquia, Colombia
[2] Facultad de Ciencias, Laboratorio de Biología Molecular y Celular, Universidad Nacional de Colombia, Medellín, Antioquia, Colombia
[3] Grupo de Investigación en Peces Neotropicales, Fundación para la Investigación y el Desarrollo Sostenible (FUNINDES), Cali, Colombia
[4] Departamento de Biología, Facultad de Ciencias, Universidad del Valle, Cali, Colombia
[*] These authors contributed equally to this work.

## ABSTRACT

The family Pseudopimelodidae is widely distributed in South America and includes 51 described species organized in seven genera. Only two of four species of the genus *Pseudopimelodus* have been recorded for the trans-Andean basins of Colombia, *Pseudopimelodus bufonius* and *P. schultzi*, whose similarity in external morphology make their identification difficult. We performed a phylogenetic analysis using a fragment of the Cytochrome C Oxidase subunit 1 gene (COI), and analyzed osteological and traditional morphometric characters to study *Pseudopimelodus* from the Colombian trans-Andean region. Results provided strong support for two clades phylogenetically related to *Pseudopimelodus,* that showed clear-cut molecular, osteological, and morphometric differences from previously described bumblebee catfishes. Based on these results, we describe two *Pseudopimelodus* species from the Magdalena-Cauca River Basin*: P. magnus* sp. nov. with 43–44 vertebrae, dorsal-fin spine with serrations on its anterior margin; lateral margin of transverse process of the fourth vertebra of the Weberian complex forming an acute angle in ventral view and *P. atricaudus* sp. nov. with 39 vertebrae, dorsal-fin spine smooth on its anterior margin and a dark, vertical band covering 3/4 of the caudal fin with base of rays and tip of caudal-fin lobes hyaline.

## INTRODUCTION

The order Siluriformes is a highly diverse group encompassing at least 3975 valid species (*Fricke, Eschmeyer & Fong, 2020*) and a growing number of new species are being described (*Shibatta, 2016*; *Ruiz, 2016*; *Shibatta & Vari, 2017*; *Shibatta, 2019*; *Tobes et al., 2020*). Within this order, Pseudopimelodidae (Bumblebee catfishes) is a small monophyletic family of Neotropical catfishes, broadly distributed across different river basins in South America, from the Atrato River in Colombia to the Río de la Plata in Argentina (*Ferraris Jr,*

Corresponding author
Edna J. Márquez,
ejmarque@unal.edu.co

*2007*) and it is considered one of the least known families among the Neotropical freshwater catfishes (*Shibatta, 2003*).

Members of Pseudopimelodidae were originally included in family Pimelodidae; furthermore, *Lundberg, Bornbusch & Mago-Leccia (1991)* demonstrated its monophyly based on several synapomorphies, and its family rank was proposed by *Shibatta (2003)*. These synapomorphies include: (1) lack of spike-like membranous wings of bone projecting beyond the palatine condyle on the lateral ethmoid; (2) a short metapterygoid, broader (at least twice) than long, deflected inward and usually subtended by a ridge or crest; (3) endopterygoid with a sharp anterolateral process and a comma-shaped ectopterygoid, both broad and loosely linked between the neurocranium, from near the lateral ethmoid-orbitosphenoid suture and the palatine; (4) third to seventh proximal radials of the dorsal fin wide and adjacent radials in full contact for their entire lengths (*Batrochoglanis*, *Cephalosilurus*, *Cruciglanis, Lophiosilurus*, and *Pseudopimelodus*) or only narrowly separated (*Microglanis*), and (5) absence of dorsal hypohyals.

In a phylogenetic analysis based on 440 characters and 87 taxa representing all extant catfish families, *Diogo, Chardon & Vandewalle (2004)* confirmed the above mentioned synapomorphies and described two additional features: (1) a markedly bifurcated mesethmoid and (2) a spoon-shaped autopalatine with a roundish, dorso-ventrally expanded posterior tip. Subsequent molecular studies based on mitochondrial and nuclear gene sequences (*Hardman, 2005*; *Sullivan, Lundberg & Hardman, 2006*; *Lundberg, Sullivan & Hardman, 2011*; *Sullivan, Muriel-Cunha & Lundberg, 2013*) showed that the family Pseudopimelodidae forms a well-supported monophyletic assemblage along with Heptapteridae, Pimelodidae, and *Phreatobius*. More recently, five additional characters were proposed for the family Pseudopimelodidae (*Shibatta & Vari, 2017*).

Until now, six genera including at least 51 species were recognized in Pseudopimelodidae: *Batrochoglanis* Gill, 1858; *Cephalosilurus* Haseman, 1911; *Cruciglanis Ortega-Lara & Lehmann, 2006*; *Lophiosilurus* Steindachner, 1876; *Microglanis* Eigenmann, 1912*; and *Pseudopimelodus* Bleeker, 1858 *Fricke, Eschmeyer & Fong, 2020*. *Shibatta & Vari (2017)* described *Rhyacoglanis* from cis-Andean South America, comprising four new species and *Pseudopimelodus pulcher* Boulenger, 1887, from the Western Amazon Basin as the type species.

So far, records in Colombia include a total of six of the seven formally recognized genera of Pseudopimelodidae and only two of the four species of *Pseudopimelodus* (*DoNascimiento et al., 2017*). Although it was thought that the species *P. bufonius* (Valenciennes, 1840) and *P. schultzi* (Dahl, 1955) had overlapping distribution ranges in the Magdalena-Cauca River Basin, in a recently published checklist (*DoNascimiento et al., 2017*), the presence of *P. bufonius* was restricted to the Amazon, Orinoco and Caribbean basins, while the species *P. schultzi* was restricted to both the Magdalena-Cauca and Caribbean basins, including the Sinú River, type locality of this species.

In addition to the difficulties to define their distribution, a phylogeographic study revealed the presence of five separate evolutionary lineages of *Pseudopimelodus* in trans- and cis-Andean rivers of Colombia (*Rangel-Medrano, Ortega-Lara & Márquez, 2020*). Thus, to test the hypothesis that undescribed species are currently classified in Colombia under

the names *P. schultzi* and *P. bufonius*, this study integrated molecular and morphological analyses, including osteology and external morphology, to address the taxonomic status of *Pseudopimelodus* species inhabiting the Magdalena-Cauca River Basin. Results identified two new species of the family Pseudopimelodidae from northwestern South America, both described in this study.

## MATERIAL AND METHODS

### Material studied

The material examined is deposited in museums and institutions as described below and institutional abbreviations used in the present study are listed in *Sabaj (2019)*.

***Pseudopimelodus bufonius***: Guyana: Upper Essequibo River Basin: Takutu River: AUM 38248, 1, 72.3 mm SL; Rupunini River: AUM 38292, cleared and stained (C&S) specimen, 58.6 mm SL; AUM 48352, 1, 74.3 mm SL; Potaro-Siparuni: AUM 45384, 1, 87.6 mm SL. Colombia: Upper Amazon River Basin: Vaupés River: IMCN 8230, 5, 137.4–221.7 mm SL. Orinoco River Basin: Upper Meta River Drainage: Negro River: IMCN 8231, 13, 118.3–191.0 mm SL. Venezuela: Orinoco River Basin: Apure River Drainage: Bocono River: MCNG 5307, 4, 103.4–123.5 mm SL.

***Pseudopimelodus schultzi***: Colombia: Sinú River: IMCN 8241, 12 (2 C&S), 174–305 mm SL.

***Pseudopimelodus mangurus***: Argentina: Upper La Plata River Basin: Paraná River: IMCN 8441, 2, 117.7–137.9 mm SL.

***Pseudopimelodus charus***: Brazil: Sao Francisco river drainage: LBP10489, 1, 123.1 mm SL; LBP 11308, 1, 154.5 mm SL.

In addition, the morphological comparisons with *Pseudopimelodus charus* were based on the figure of Valenciennes published in *Mees (1974)*.

### Phylogenetic analyses

Phylogenetic analysis was conducted with MrBayes (MB) v3.2 (*Ronquist et al., 2012*) using COI haplotype sequences from *Pseudopimelodus* specimens (Table 1), belonging to lineages 1 and 5 described by *Rangel-Medrano, Ortega-Lara & Márquez (2020)*. Haplotypes were compared with GenBank COI sequences of remaining members of the family Pseudopimelodidae (Table 1), using HKY+G as the best-fit evolutionary model estimated in IQ-TREE software (*Kalyaanamoorthy et al., 2017*). Two Pimelodid species (*Pimelodus yuma* and *Pseudoplatystoma magdaleniatum*) were used as outgroups. Chain parameters included two independent Markov Chain Monte Carlo (MCMC) iterations for 20 million generations sampled every 1,000 generations, discarding the first 25% sampled generations as burn-in; remaining parameters were left as default. Convergence of the MCMC was assessed based on the Potential Scale Reduction Factor which should approach 1.0 as runs converge and considering the standard deviation of split frequencies which should approach 0. Trees were summarized according to their estimated posterior probability to produce a consensus tree, using the same burn-in as the MCMC. The final tree was visualized with the program FigTree v1.4.2 (*Rambaut, 2014*). Nodes were considered well supported with posterior probabilities $\geq$ 0.95 (*Wilcox et al., 2002*). Finally, following the

**Table 1** List of COI sequences of species of Pimelodoidea used for phylogenetic analysis.

| Species | Geography | GenBank accession | Sample size |
|---|---|---|---|
| *Pseudopimelodus atricaudus* H1[a] | San Jorge River | MH553571 | 1 |
| *Pseudopimelodus atricaudus* H8[a] | Cauca River lower sector | MH800619–MH800632, MH800634–MH800639 | 20 |
| *Pseudopimelodus atricaudus* H11[a] | Cauca River lower sector | MH553581 | 1 |
| *Pseudopimelodus atricaudus* H10[a] | Magdalena River lower sector | MH553580 | 1 |
| *Pseudopimelodus magnus* H13[a] | Cauca River middle sector | MH553583 | 1 |
| *Pseudopimelodus magnus* H14[a] | Cauca River middle sector | MH553584 | 1 |
| *Pseudopimelodus magnus* H16[a] | Cauca River middle sector | MH553586 | 1 |
| *Pseudopimelodus magnus* H17[a] | Magdalena River upper sector | MH800809–MH800812 | 4 |
| *Pseudopimelodus magnus* H18[a] | Cauca River middle sector | MH800711–MH800715 | 5 |
| *Pseudopimelodus magnus* H19[a] | Cauca River middle sector | MH553589 | 1 |
| *Pseudopimelodus magnus* H20[a] | Cauca River middle sector | MH553590 | 1 |
| *Pseudopimelodus magnus* H23[a] | Cauca River upper sector | MH553593 | 1 |
| *Pseudopimelodus magnus* H24[a] | Cauca River upper sector | MH553594 | 1 |
| *Pseudopimelodus schultzi* H25[a] | Sinú River | MH553595 | 1 |
| *Pseudopimelodus schultzi* H26[a] | Sinú River | MH553596 | 1 |
| *Pseudopimelodus bufonius* H27[a] | Orotoy River | MH553597 | 1 |
| *Pseudopimelodus bufonius* H28[a] | Vaupés River | MH553598 | 1 |
| *Pseudopimelodus bufonius* H29[a] | Orteguaza River | MH553599, MH800832 | 2 |
| *Pseudopimelodus mangurus*[b] | Paranapanema River | EU179816 | 1 |
| *Pseudopimelodus charus*[b] | São Francisco River | EU179815 | 1 |
| *Cruciglanis* sp. AOL024[a] | Mira River | MH553609 | 1 |
| *Cruciglanis pacifici* AOL094[a] | Anchicayá River | MH553607 | 1 |
| *Rhyacoglanis annulatus* AOL097-98[a] | Meta River | MH553605–MH553606 | 2 |
| *Rhyacoglanis pulcher*[b] | — | EU179812 | 1 |
| *Cephalosilurus apurensis*[b] | Orinoco River | EU179818 | 1 |
| *Batrochoglanis raninus*[b] | Aquarium | EU179809 | 1 |
| *Lophiosilurus alexandri*[c] | São Francisco River | HM405152 | 1 |
| *Microglanis* sp. AOL095[a] | Acacias River | MH553604 | 1 |
| *Pimelodus yuma*[a] | Cauca River lower sector | MH553610 | 1 |
| *Pseudoplatystoma magdaleniatum*[a] | Cauca River lower sector | MH553611 | 1 |

**Notes.**
[a] *Rangel-Medrano, Ortega-Lara & Márquez (2020)*.
[b] C Oliveira, 2018, pers. comm.
[c] *De Carvalho et al. (2011)*

DNA barcoding of freshwater fishes (*Hubert et al., 2008*), the pair-wise divergences of *Pseudopimelodus* haplotype sequences was estimated using the Kimura 2 parameter model in MEGA v6.06 (*Tamura et al., 2013*).

## Morphological analyses

Measurements followed standard procedures of *Shibatta & Vari (2017)* using dial calipers to 0.1 mm on the left side of specimens. Standard length is expressed in mm. Except for subunits of the head which are expressed as percentages of head length, all measurements are expressed as percentages of standard length. Counts were made on the left side of the

body when possible. Counts of dorsal, pectoral, pelvic, and anal fin rays, as well as principal and procurrent caudal-fin rays were taken from dried skeletons (DS), cleaned using dermestid beetles, C&S specimens following *Taylor & Van Dyke (1985)* and radiographs (RX) of paratypes. Vertebral counts include the five fused vertebrae of the Weberian apparatus and one single element of the compound caudal centrum (pleural 1 + ural 1 centrum). Length of the posterior process of cleithrum was measured from dorsal origin to its posterior end. Width of the pectoral-fin spine was measured at its base. Length of the Weberian complex was measured in ventral view, longitudinally from anterior to posterior mesial ends of centrum and was related to the length of the neurocranium, which was measured from the mesial anterior margin of the mesethmoid to posterior margin of the basioccipital. Osteological nomenclature follows *Arratia (2003a)* and *Arratia (2003b)*.

The electronic version of this article in Portable Document Format (PDF) will represent a published work according to the International Commission on Zoological Nomenclature (ICZN), and hence the new names contained in the electronic version are effectively published under that Code from the electronic edition alone. This published work and the nomenclatural acts it contains have been registered in ZooBank, the online registration system for the ICZN. The ZooBank LSIDs (Life Science Identifiers) can be resolved and the associated information viewed through any standard web browser by appending the LSID to the prefix http://zoobank.org/. The LSID for this publication is: urn:lsid:zoobank.org:pub: 8B78D766-07A3-47A8-A12C-F55958703ACB. The online version of this work is archived and available from the following digital repositories: PeerJ, PubMed Central and CLOCKSS.

# RESULTS

## Molecular analysis

Phylogenetic tree based on Bayesian Inference (Fig. 1), showed well-supported clades corresponding to samples here proposed as new species (*Pseudopimelodus magnus* and *P. atricaudus*). *Pseudopimelodus magnus* was recovered sister to *P. schultzi* from the Sinú River (H25 and H26) and both clades along with *P. bufonius* from the Orinoco, appear as the sister-clade to *P. bufonius* from Amazonian rivers, *P. mangurus,* and *P. charus*. In contrast, *P. atricaudus* is sister to the remaining *Pseudopimelodus* species analyzed.

Kimura 2 parameters genetic distance was relatively small for *P. atricaudus* (0.003–0.006), *P. magnus* (0.003–0.020), and *P. schultzi* (0.020), whereas was larger between *P. bufonius* samples (0.011–0.068). The smallest Kimura 2 parameters genetic distance was observed between *P. bufonius* from Orteguaza and *P. mangurus* (0.026), whereas the largest genetic distance was observed between *P. schultzi* and *P. atricaudus* (0.113).

***Pseudopimelodus magnus,*** **sp. nov.** (Fig. 2)

*Pseudopimelodus bufonius* [not Valenciennes, 1840]. —*Steindachner, 1880*: 59 [Cauca River; description]. —*Miles, 1947*: 64 [Magdalena River Basin; check list]. —*Dahl, 1971*: 54 [Magdalena River Basin; check list]. —*Maldonado-Ocampo et al., 2005*: 159 fig. 150 [Magdalena-Cauca River basin; check list].

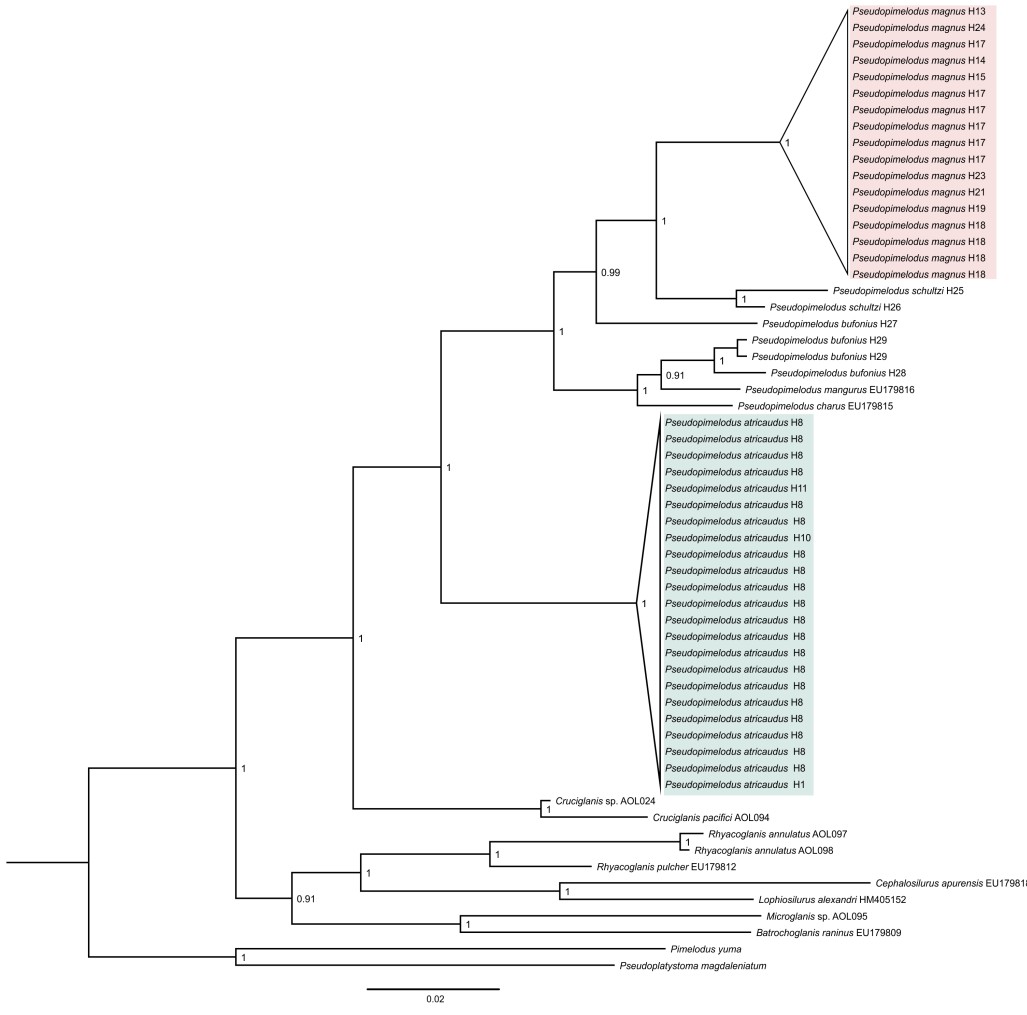

**Figure 1** Bayesian phylogenetic tree based on partial sequences of COI gene, showing the phylogenetic position of *Pseudopimelodus magnus* and *P. atricaudus* within Pseudopimelodidae.

*Pseudopimelodus schultzi* [not Dahl, 1955]. —*Ortega-Lara et al., 2006*: 49 [upper Cauca Basin; check list]. —*Ortega-Lara, Rivas & Rincón, 2011*: 447 [fishery: Magdalena River Basin].

*Pseudopimelodus zungaro* [not Humboldt, 1833]. —*Eigenmann & Eigenmann, 1890*: 112 [Goiás, Brazil]. —*Eigenmann, 1922*: 32 [in part, Magdalena-Cauca River Basin; identification key].

**Holotype.** CIUA 5142, 750 mm SL, Colombia, Antioquia, Magdalena River Basin, Cauca River in Venecia, 6°39′47.6″N, 75°50′9″W, Feb 2018, Restrepo-Gómez A.M.

**Paratypes.** ANDES I210, 2, 232.9–248.5 mm SL, Colombia, Antioquia, Magdalena River Basin, Cauca River in Venecia, 6°39′47.6″N, 75°50′9.5″W, Feb 2016, Restrepo-Gómez A.M. CIUA 5163, 1, 531.7 mm SL, Colombia, Antioquia, Magdalena River Basin, Cauca River in Venecia, 6°39′47.6″N, 75°50′9″W, Nov 2014, Olaya G. CP-UCO 3860, 1, 217.7 mm SL, Colombia, Antioquia, Magdalena River Basin, Cauca River in Venecia, 6°39′47.6″N,

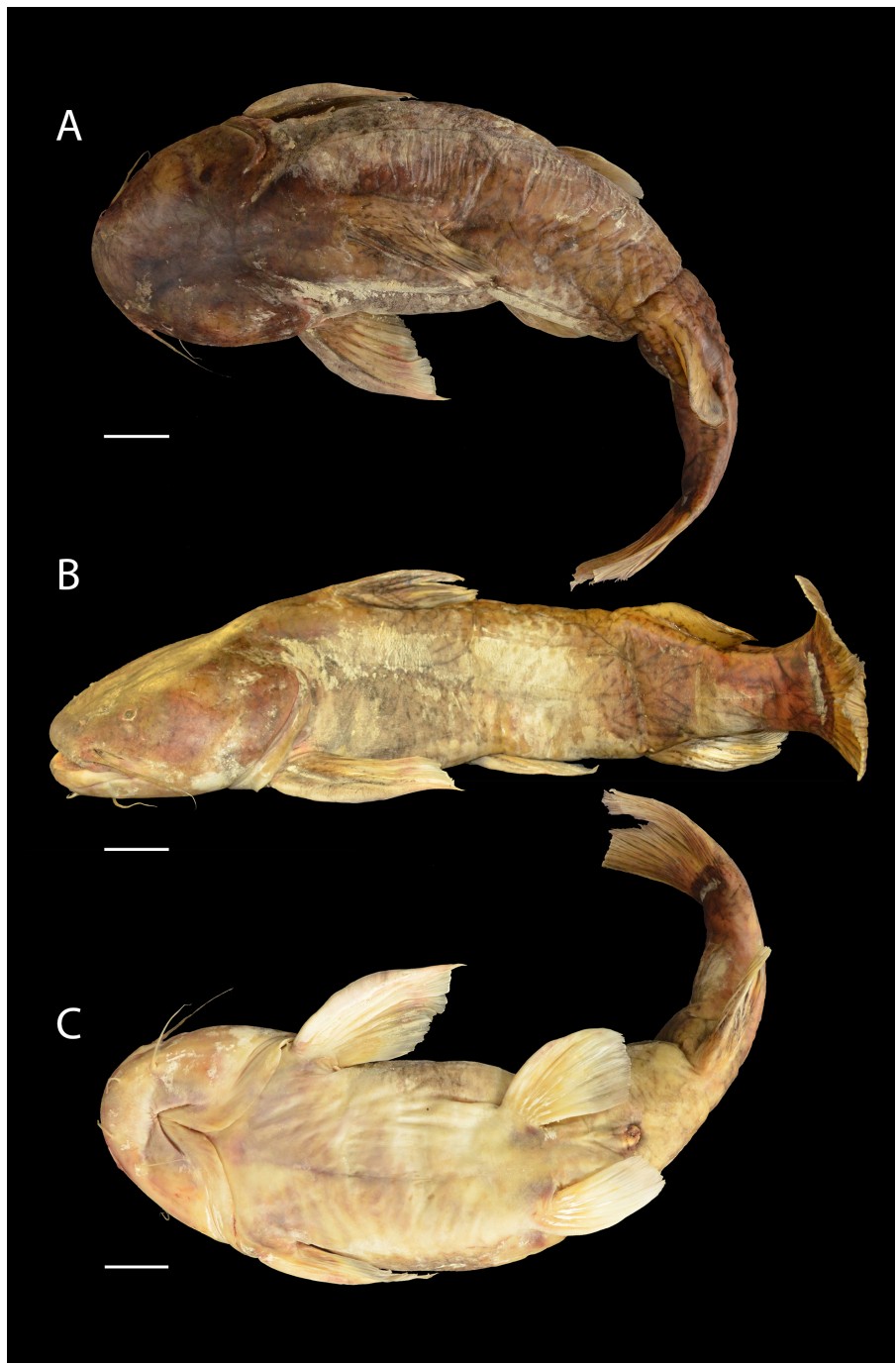

**Figure 2** *Pseudopimelodus magnus,* **holotype, CIUA 5142, 750 mm SL, Colombia, Antioquia, Magdalena River basin, Cauca River in Venecia.** (A, B, C): Dorsal, lateral, and ventral view, respectively. Scale bar = five cm. Photography: Giovany Olaya Betancur. Photographic edition: Mauricio Urrego Tobón.

75°50′9″W, Nov 2015, Restrepo-Gómez A.M. CP-UCO 3861, 1 230.1 mm SL, Colombia, Bolívar, Magdalena River Basin, Cauca River in Achí, 8°34′8.1″N, 74°33′10.9″W, Oct 2015, Restrepo-Gómez A.M. CP-UCO 3862, 1, 177.9 mm SL, Colombia, Antioquia, Magdalena River Basin, Cauca River in Venecia, 6°39′47.6″N, 75°50′9″W, Feb 2016, Restrepo-Gómez A.M. ICN-MHN 24386, 1, 326.84 mm SL, Colombia, Antioquia, Magdalena River Basin, Cauca River in Venecia, 6°39′47.6″N, 75°50′9.5″W, Dec 2015, Restrepo-Gómez A.M. IMCN 0060, 1, 106 mm SL, Colombia, Valle del Cauca, Upper Cauca River Basin, Cauca River in Tuluá town, *ca* 4°6′55″N, 76°17′47″W, Nov 1995, Victoria P. IMCN 0285, 1, 228 mm SL, Colombia, Cauca, Upper Cauca River Basin, Ovejas River in Suárez, *ca*. 2°57′41″N, 76°40′39″W, Nov 1995, Usma S. IMCN 2349, 2 DS, 157.4–408 mm SL, Colombia, Cauca, Upper Cauca River Basin, Cauca River in El Hormiguero, Municipality of Puerto Tejada, *ca*. 3°18′05″N, 76°28′39″W, 5 Nov 2002, Ortega-Lara A. IMCN 3885, 2, 57.3–62.5 mm SL, Colombia, Valle del Cauca, Upper Cauca River Basin, La Vieja River in Cartago town, *ca*. 4°45′48″N, 75°55′27″W, 31 Aug 2004, Ortega-Lara A. IMCN 8234, 6, 1 DS, 268.1–340 mm SL, Colombia, Huila, Magdalena River Basin, Magdalena River in El Quimbo dam, Municipality of Garzón, *ca*. 2°11′05″N, 75°39′44″W, 29 Aug 2015, Ortega-Lara A. IMCN 8239, 1 C&S, not measured, Colombia, Cauca, Cauca River Basin, Cauca River in El Hormiguero, Municipality of Puerto tejada, *ca*. 3°18′25″N, 76°28′26″W, 1 Mar 2003, Ortega-Lara A. IMCN 8265, 1 DS, 213.7 mm SL, Colombia, Antioquia, Magdalena River Basin, Cauca River in Venecia, 6°39″47.6″N, 75°50′9″W, Feb 2016, Restrepo-Gómez A.M. IMCN 8267, 1 DS, 193.3 mm SL, Colombia, Antioquia, Magdalena River Basin, Cauca River in Venecia, 6°39′47.61″N, 75°50′9″W, Feb 2016, Restrepo-Gómez A.M. IMCN 8269, 1 RX, 217 mm SL, Colombia, Antioquia, Magdalena River Basin, Cauca River, Valdivia stream in Puerto Valdivia, 7°17′17.7″N, 75°23′32.1″W, Feb 2015, Olaya G. IMCN 8270, 1 RX, 294.2 mm SL, Colombia, Antioquia, Magdalena River Basin, Cauca River in Venecia, 6°39′47.61″N, 75°50′9″W, Apr 2015, Olaya G. IMCN 8939, 16, 171–475 mm SL, Colombia, Cauca, Upper Cauca River Basin, Cauca River in El Hormiguero, Municipality of Puerto Tejada, *ca*. 3°18′05″N, 76°28′39″W, 22 Dic 2011, Ortega-Lara A.

### Non-type material

CIUA 405, 53.8 mm SL Colombia, Valle del Cauca, Cauca River Basin, La Vieja River, 4°41′9.8″N, 75°50′57.7″W, Aug 2006, Ochoa L., Montoya A.F. CIUA 512, 160 mm SL, Colombia, Antioquia, Cauca River Basin, Cauca River in Bolombolo, 5°58′01.3″N, 75°50′26.5″W, Aug 2006, Ochoa L., Montoya A.F.

**Diagnosis.** *Pseudopimelodus magnus* differs from its congeners by the total number of vertebrae (43–44, Fig. 3A *vs.* 39 in *P. atricaudus* Fig. 3B; 38 in *P. bufonius*; 38–40 in *P. mangurus*; 41 in *P. schultzi*) and by the deeply acute notch on lateral margins of transverse process of the fourth vertebra of the Weberian complex (Fig. 4A *vs.* shallowly concave in *P. atricaudus*, *P. bufonius*, *P. mangurus*, and *P. schultzi*). It differs from *P. atricaudus* by the angle of < 90° formed by the medial junction of the posterior arm of the transverse process of the fourth vertebra (Figs. 4 and 5 *vs.* angle almost right). It differs from *P. atricaudus* and *P. mangurus* by length ratio of Weberian complex and neurocranium (32.1–36.3%, Fig. 5 *vs.* 46.9–55.5 in *P. atricaudus* and 42.7% in *P. mangurus*). It differs from *P. atricaudus*, *P.

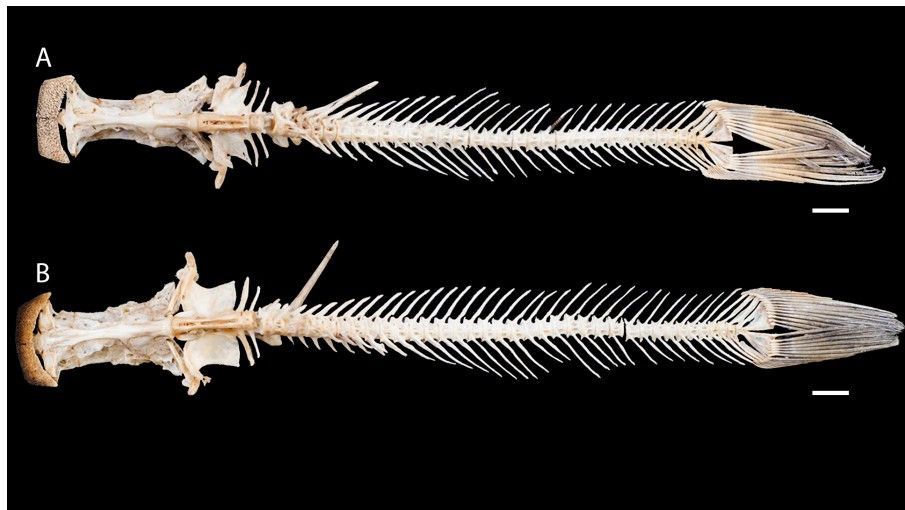

**Figure 3** **Dried skeletons of *Pseudopimelodus magnus*, paratype, IMCN 8265, 213.7 mm SL (A) and *P. atricaudus*, paratype, IMCN 8266, 203.5 mm SL (B).** Scale bar = one cm. Photography: Anderson Cardona Ruiz. Photographic edition: Mauricio Urrego Tobón.

*bufonius*, and *P. mangurus* by having shorter maxillary barbels, not surpassing the opercular margin (*vs*. reaching). Additionally, *P. magnus* differs from the remaining species, except *P. atricaudus*, by the length of the posterior process of cleithrum (1.32–1.59 times the wide of the pectoral-fin spine base *vs*. 0.57–0.75 in *P. bufonius*, 2.22–2.76 in *P. schultzi*, 2.24–2.30 in *P. mangurus*). It differs from *P. bufonius*, *P. charus*, and *P. mangurus* by having a heart-shaped gas bladder (*vs*. dumbbell-shaped bladder; Fig. 6; (*Shibatta & Vari, 2017*). It differs from *P. atricaudus* by having the anterior margin of the dorsal-fin spine serrated (*vs*. smooth). *Pseudopimelodus magnus* differs from congeners except *P. bufonius* and *P. schultzi* by having a narrow vertical dark band along the center of caudal fin, although in some specimens is hardly visible (*vs*. broad vertical dark band covering $\leq \frac{1}{2}$ caudal fin in *P. charus* and *P. mangurus* or $\frac{3}{4}$ of the caudal fin with base of rays and tip of caudal-fin lobes hyaline in *P. atricaudus*).

**Description**. Morphometric data in Table 2. Body depressed from snout tip to dorsal-fin origin; progressively compressed towards caudal-fin base. Snout rounded in dorsal view. Head trapezoidal, depressed, slightly longer than wide. Head covered by thick skin hiding fontanel and cranial roof bones. Eye small, covered by skin and positioned latero-dorsally. Prognathous jaw. Teeth small and villiform; premaxilla laterally projected backwards and reaching lateral process of lateral ethmoid. Anterior nostril tubular located lateroposteriorly to maxillary barbel base (Fig. 2). Posterior nostril equidistant from anterior nostril and eye. Maxillary barbel not reaching opercular margin. Mental barbel anteriorly inserted to gular apex. Inner mental barbel reaching gular apex. Outer mental barbel not reaching branchial opening. Gular fold V-shaped, with conspicuously pointed apex (Fig. 2). Branchiostegal membrane free from isthmus. Posterior process of vomer bifurcated. Posterior process of cleithrum triangular, its length 1.25–1.59 times width of pectoral-spine base. Vomer T-shaped, in contact with parasphenoid, mesethmoid, and lateral ethmoid. Posterior region

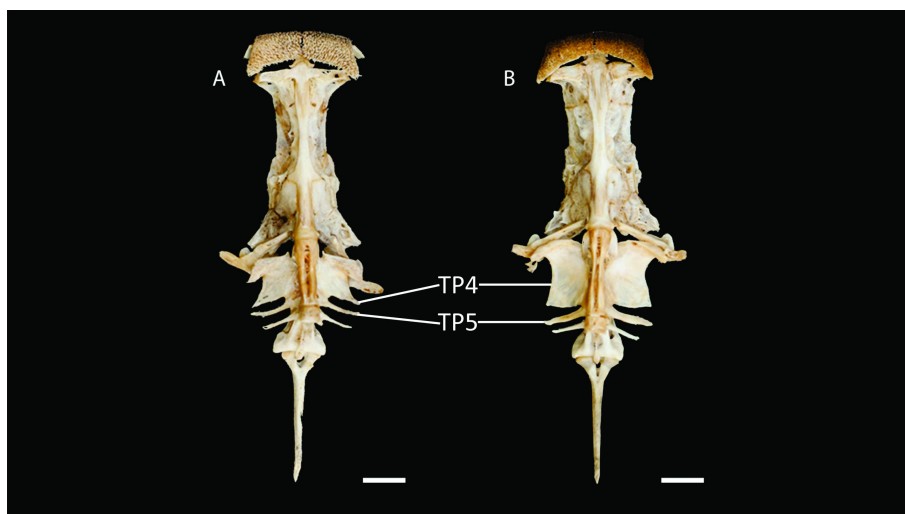

**Figure 4** **Neurocranium and Weberian complex in ventral view of *Pseudopimelodus magnus,* paratype, IMCN 8265, 213.7 mm SL (A) and *P. atricaudus,* paratype, IMCN 8266, 203.5 mm SL (B).** TP4: transverse process of the fourth vertebra, TP5: transverse process of the fifth vertebra. Scale bar: one cm. Photography: Anderson Cardona Ruiz. Photographic edition: Mauricio Urrego Tobón.

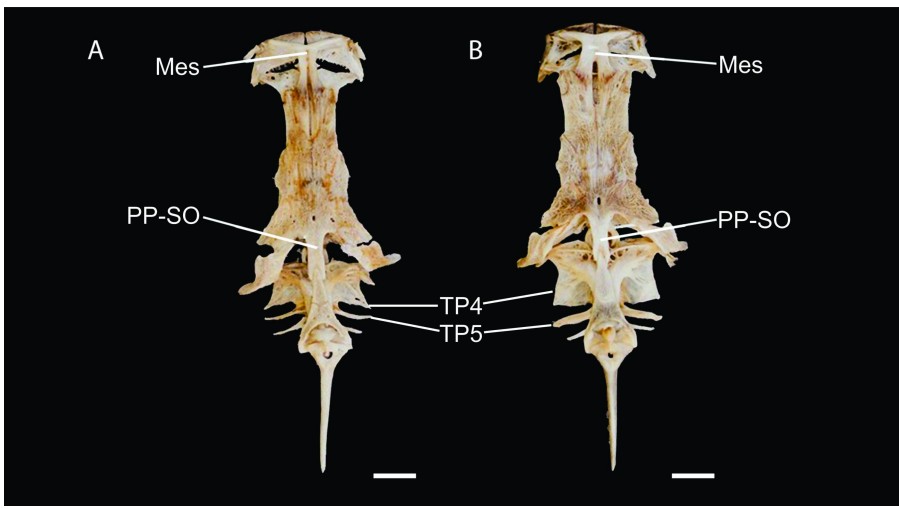

**Figure 5** **Neurocranium and Weberian complex in dorsal view of *Pseudopimelodus magnus,* paratype, IMCN 8265, 213.7 mm SL (A) and *P. atricaudus,* paratype, IMCN 8266, 203.5 mm SL (B).** Mes: mesethmoid; PP-SO: parieto-supraoccipital; TP4: transverse process of the fourth vertebra; TP5: transverse process of the fifth vertebra. Scale bar: one cm. Photography: Anderson Cardona Ruiz. Photographic edition: Mauricio Urrego Tobón.

of mesethmoid wider than base of parieto-supraoccipital process (Fig. 5A). Transverse process of the fourth vertebra of Weberian complex forming an angle > 90° between anterior and lateral margins and < 90° between lateral and posterior margins, in ventral

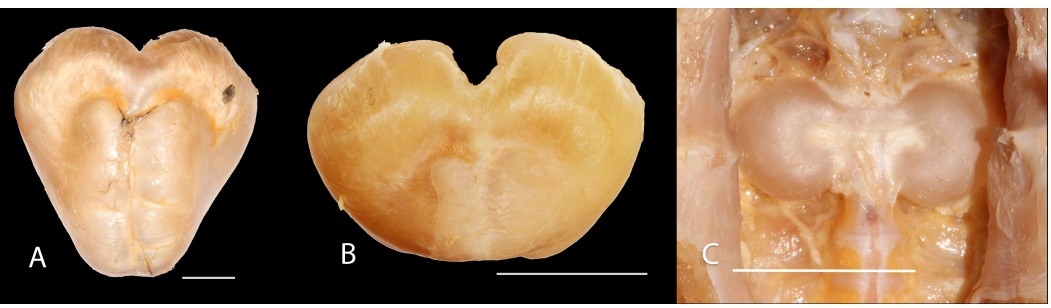

**Figure 6** Gas bladder in ventral view of *Pseudopimelodus magnus*, IMCN 8939, 361 mm SL (A), *P. atricaudus*, IMCN 4196, 104.1 mm SL (B) and *P. bufonius*, Vaupés IMCN 8230, 137.4 mm SL (C).

view (Fig. 4). Length of Weberian complex in relation to length of neurocranium 32.2–36.3%. Anterior fontanel elongated, not reaching transverse line through opening of infraorbital sensory canal in sphenotic. Posterior fontanel small and oval-shaped, located at center of parieto-supraoccipital (Fig. 5A). Parieto-supraoccipital process with bifurcated tip in contact with supraneural (Fig. 5A). Heart-shaped gas bladder (Fig. 6). Lateral line complete, reaching caudal-fin base. Number of total vertebrae 43–44. Ribs 13. Axillary pore present. Dorsal-fin origin at anterior third of body. Posterior margin of fin convex. Dorsal-fin spine strongly ossified with distal end pointed, shorter than next branched ray and with serrations on anterior margin. Dorsal-fin rays I, 6. Adipose-fin origin anterior to anal-fin origin. Pectoral-fin spine serrated, covered by skin; anterior margin with 18–41 serrations of similar size and posterior margin with 17–27 larger serrations. Posterior edge of pectoral fin convex. Axillary pore present. Pectoral-fin rays I, 7. Posterior margin of pelvic fin rounded; origin inserted behind posterior end of dorsal-fin base. Pelvic-fin rays i, 5. Anal fin with posterior edge rounded and inserted below 20th vertebra. Anal-fin rays iii, 7. Caudal fin bifurcated, with pointed lobes, upper lobe slightly narrower and longer than ventral lobe. Caudal-fin rays i, 8, 7, i; dorsal procurrent rays 14; ventral procurrent rays 14.

**Color in life and alcohol**. Background of body of variable color from yellow to light brown with four vertical dark bands (Fig. 7). First predorsal band partially conjoined with subdorsal band at level of upper corner of branchial margin. Subdorsal band connected dorsally with sub-adipose band, almost reaching anal-fin base. Dark band on caudal-fin base, completely fused or separated from sub-adipose band in dorsal region. In some specimens, bands less evident and faint. Region between bands with small, scattered and irregularly-shaped spots. Body covered by bright yellow mucus. Pectoral, ventral, and dorsal fins reddish with two transverse dark bands, at base and last-third of each fin. Adipose-fin of same color as body, occasionally covered by sub-adipose band in middle region. Caudal fin pale brown with or without narrow dark band along the center of caudal fin. Color in alcohol (Fig. 2) similar to live coloration, except regions lacking superficial mucus, becoming darker (grey to brown).

**Distribution.** Magdalena River, from upper sector at El Quimbo dam to the confluence with the Gualí River in Honda town. Cauca River throughout the entire basin (Fig. 8).

**Table 2  Morphometric data of *Pseudopimelodus magnus* (32 specimens).**

|  | Holotype | Max | Min | Mean | SD |
|---|---|---|---|---|---|
| Standard length (mm) | 750 | 531.7 | 168 | 279.4 | 86 |
| **Percentages of standard length** | | | | | |
| Head length | 23.3 | 32.1 | 23.3 | 30.0 | 1.4 |
| Pre-dorsal distance | 30.0 | 36.8 | 30.0 | 34.5 | 1.2 |
| Pre-pectoral distance | 24.0 | 27.6 | 23.7 | 25.2 | 1.0 |
| Pre-pelvic distance | 41.3 | 58.8 | 41.3 | 53.2 | 2.9 |
| Distance between pectoral-fin origin and dorsal-fin origin | 21.3 | 25.2 | 21.3 | 22.8 | 1.0 |
| Distance between pectoral-fin origin and pelvic-fin origin | 22.0 | 35.7 | 22.0 | 31.4 | 2.5 |
| Distance between dorsal-fin origin and pelvic-fin origin | 26.7 | 30.7 | 23.8 | 26.7 | 1.8 |
| Dorsal-fin base length | 8.4 | 14.1 | 8.4 | 12.5 | 1.0 |
| Distance between adipose-fin origin and pelvic-fin origin | 24.7 | 33.9 | 24.7 | 28.8 | 2.0 |
| Distance between pelvic-fin origin and anal-fin origin | 19.7 | 27.9 | 19.7 | 25.4 | 1.6 |
| Distance between anal-fin origin and adipose-fin origin | 19.3 | 19.3 | 13.1 | 15.7 | 1.4 |
| Adipose-fin base length | 9.8 | 14.7 | 9.8 | 13.0 | 1.2 |
| Anal-fin base length | 13.6 | 13.6 | 7.9 | 9.8 | 1.1 |
| Pectoral-fin base length | 18.7 | 27.8 | 18.7 | 25.5 | 1.7 |
| Pelvic-fin base length | 14.0 | 19.2 | 11.8 | 14.4 | 1.7 |
| **Percentages of head length** | | | | | |
| Orbital diameter | 5.6 | 9.1 | 4.6 | 6.0 | 1.1 |
| Snout length | 41.7 | 41.7 | 24.5 | 33.6 | 2.8 |
| Distance between maxillary barbels | 53.1 | 53.1 | 39.6 | 46.2 | 2.6 |
| Distance between anterior most mesial point of snout and left anterior nostril | 11.6 | 18.8 | 9.3 | 14.0 | 2.6 |
| Distance between maxillary barbel and eye | 14.0 | 19.1 | 13.3 | 15.8 | 1.3 |
| Distance between anterior nostrils | 29.2 | 29.2 | 19.7 | 24.0 | 2.0 |
| Distance between posterior nostrils | 34.5 | 34.5 | 23.0 | 27.2 | 2.3 |
| Distance between anterior and posterior nostrils | 7.0 | 8.9 | 5.0 | 7.2 | 0.9 |
| Distance between posterior nostril to eye | 9.0 | 9.0 | 4.4 | 7.1 | 1.0 |
| Interorbital distance | 49.6 | 49.6 | 34.3 | 41.1 | 2.7 |
| Mouth width | 73.2 | 82.7 | 60.1 | 71.9 | 5.9 |
| Distance between outer mental barbels | 42.4 | 46.9 | 34.0 | 39.1 | 2.9 |
| Distance between inner mental barbels | 23.3 | 24.5 | 16.3 | 21.0 | 2.2 |

**Etymology.** The specific name *magnus* is from the Latin, meaning "great" and refers to the fact that is the largest species of *Pseudopimelodus* described so far (see *Shibatta, 2003*; *Ortega-Lara & Lehmann, 2006*).

***Pseudopimelodus atricaudus*, sp. nov.** (Fig. 9)

*Pseudopimelodus schultzi* [not Dahl, 1955]. —*Ortega-Lara & Lehmann, 2006*: 155 [Magdalena Basin; comparative material]. —*Mojica et al., 2006*: 33 [middle Magdalena basin; check list]. —*Villa-Navarro et al., 2006*: 15 [upper Magdalena basin; check list]. *Pseudopimelodus* cf. *bufonius* [not Valenciennes, 1840]: —*Jiménez-Segura & Ortega-Lara, 2011*: 545 [Magdalena River Basin; fisheries].

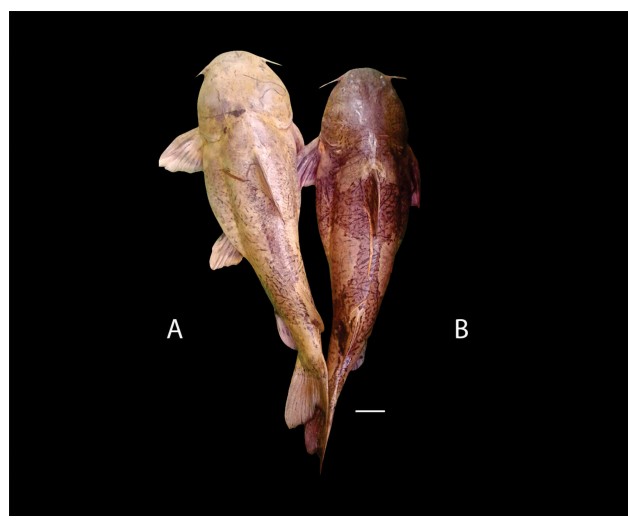

**Figure 7** **Live individuals of *Pseudopimelodus magnus* from Colombia, Antioquia, Magdalena River basin, Cauca River in Venecia.** (A) Specimen not collected, showing alternative coloration pattern. (B) holotype. Scale bar: five cm. Photography: Ana M. Restrepo Gómez. Photographic edition: Mauricio Urrego Tobón.

**Holotype.** CIUA 5141, 143.7 mm SL, Colombia, Sucre, Magdalena River Basin, Cauca River in Guaranda, 8°19′04.6″N, 74°31′56.9″W, Feb 2018, Restrepo-Gómez A.M.
**Paratypes.** ANDES I211, 1, 185.3 mm SL, Colombia, Antioquia, Magdalena River Basin, Cauca River in Barrio Chino, Caucasia, 8°00′35.2″N, 75°00′0″W, Dec 2015, Restrepo-Gómez A.M. ANDES I212, 1, 203.5 mm SL, Colombia, Antioquia, Magdalena River Basin, Cauca River in Paloma, Caucasia, 7°59′31.7″N, 74°58′35.6″W, Dec 2015, Restrepo-Gómez A.M. CIUA 5159, 1, 241.0 mm SL, Colombia, Antioquia, Magdalena River Basin, Cauca River in Barrio Chino, Caucasia, 08°00′35.3″N, 75°0′00″W, Dec 2015, Restrepo-Gómez A.M. CIUA 5160, 1, 190.0 mm SL, Colombia, Antioquia, Magdalena River Basin, Cauca River in La Ilusión, Caucasia, 8°1′50″N, 75°0′0.9″W, Dec 2015, Restrepo-Gómez A.M. CIUA 5161, 1, 205.0 mm SL, Colombia, Bolívar, Magdalena River Basin, Cauca River in Punta Cartagena, Pinillos, 8°53′37.3″N, 74°28′28.4″W, Feb 2014, Olaya G. CIUA 5162, 1, 250.0 mm SL, Colombia, Antioquia, Magdalena River Basin, Cauca River in Venecia, 6°39′47.6″N, 75°50′9″W, Dec 2015, Olaya G. CP-UCO 3857, 3, 131.8 –198.9 mm SL, Colombia, Antioquia, Magdalena River Basin, Cauca River in Barrio Chino, Caucasia, 8°0′35.3″N, 75°0′0″W, Dec 2015, Restrepo-Gómez A.M. CP-UCO 3858, 3, 153.8–192.6 mm SL, Colombia, Bolívar, Magdalena River Basin, Cauca River in Punta Cartagena, Pinillos, 8°53′37.3″N, 74°28′28.4″W, Jan 2015, Olaya G. CP-UCO 3859, 1, 109.0 mm SL, Colombia, Bolívar, Magdalena River Basin, Caribona River in Montecristo, 8°19′6.1″N, 74°31′7.9″W, Dec 2015, Restrepo-Gómez A.M. CZUT-IC 1785, 2, 136–178 mm SL, Colombia, Tolima, Magdalena River Basin, Magdalena River in Honda, 5°14′05″N, 74°43′43″W, 15 Jan 2005. ICN-MHN 24387, 1, 147.9 mm SL, Colombia, Bolívar, Magdalena River Basin, Cauca River in Punta Cartagena, Pinillos, 8°53′37.3″N, 74°28′28.4″W, Jan 2015, Olaya G. IMCN 0324, 2, 139.7–185.3 mm SL, Colombia, Santander, Magdalena River Basin, Magdalena

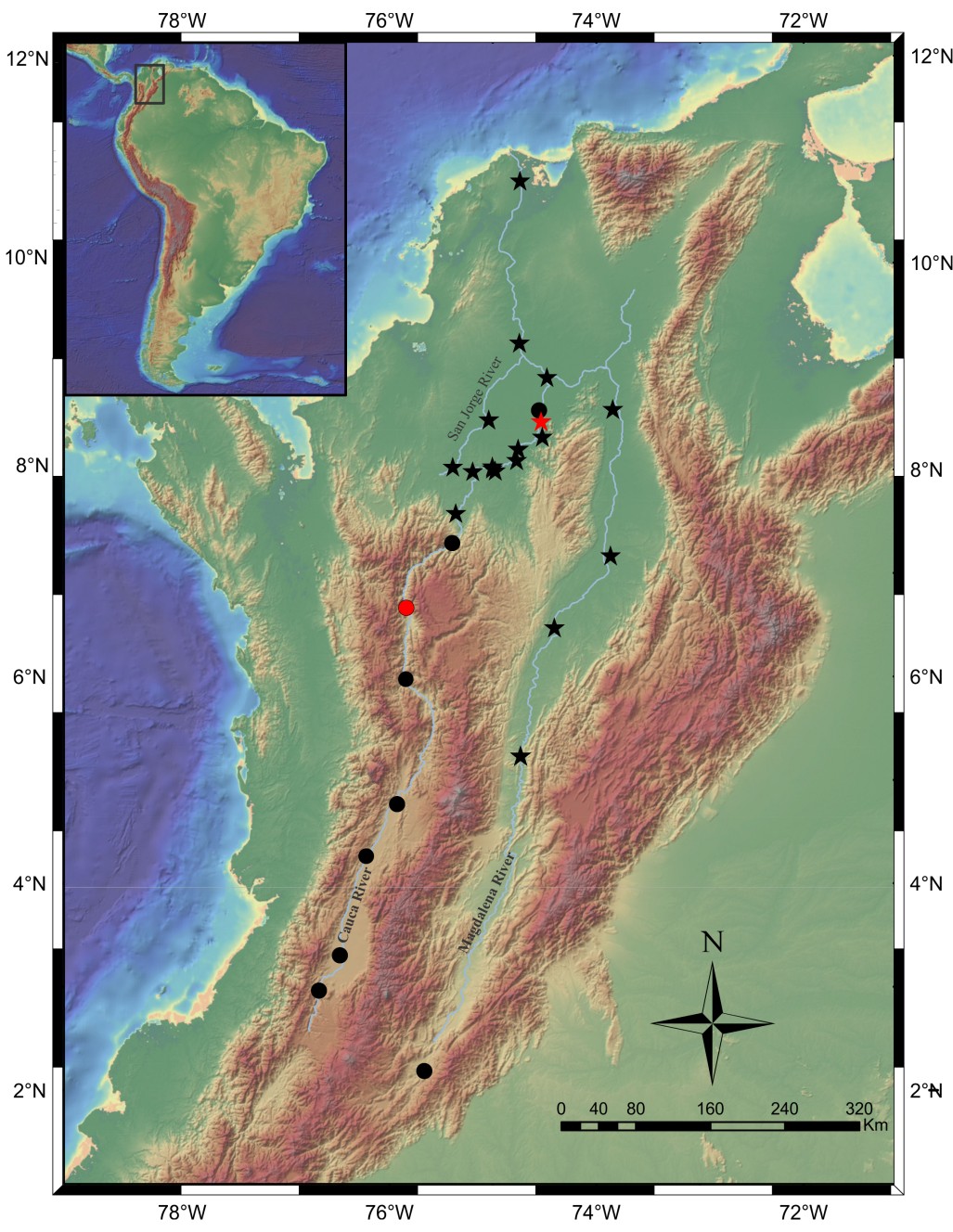

**Figure 8** **Map of the Magdalena-Cauca River Basin showing the geographic distribution of** ***Pseudopimelodus magnus*** **(circles) and** ***P. atricaudus*** **(stars).** Red symbols denote type locality, black symbols denote additional collection sites. Map image layer by NOAA National Centers for Environmental Information (NCEI).

River in Puerto Wilches, *ca.* 7°19′39″N, 73°54′41″W, 27 Jun 1996, Aldana J. IMCN 4196, 1 C&S, 6, 90.08–210.3 mm SL, Colombia, Bolivar, Magdalena River Basin, Magdalena River in Hatillo de la Loba, Oct 2006, Ardila C. IMCN 8232, 3, 153.8–180.4 mm SL, Colombia,

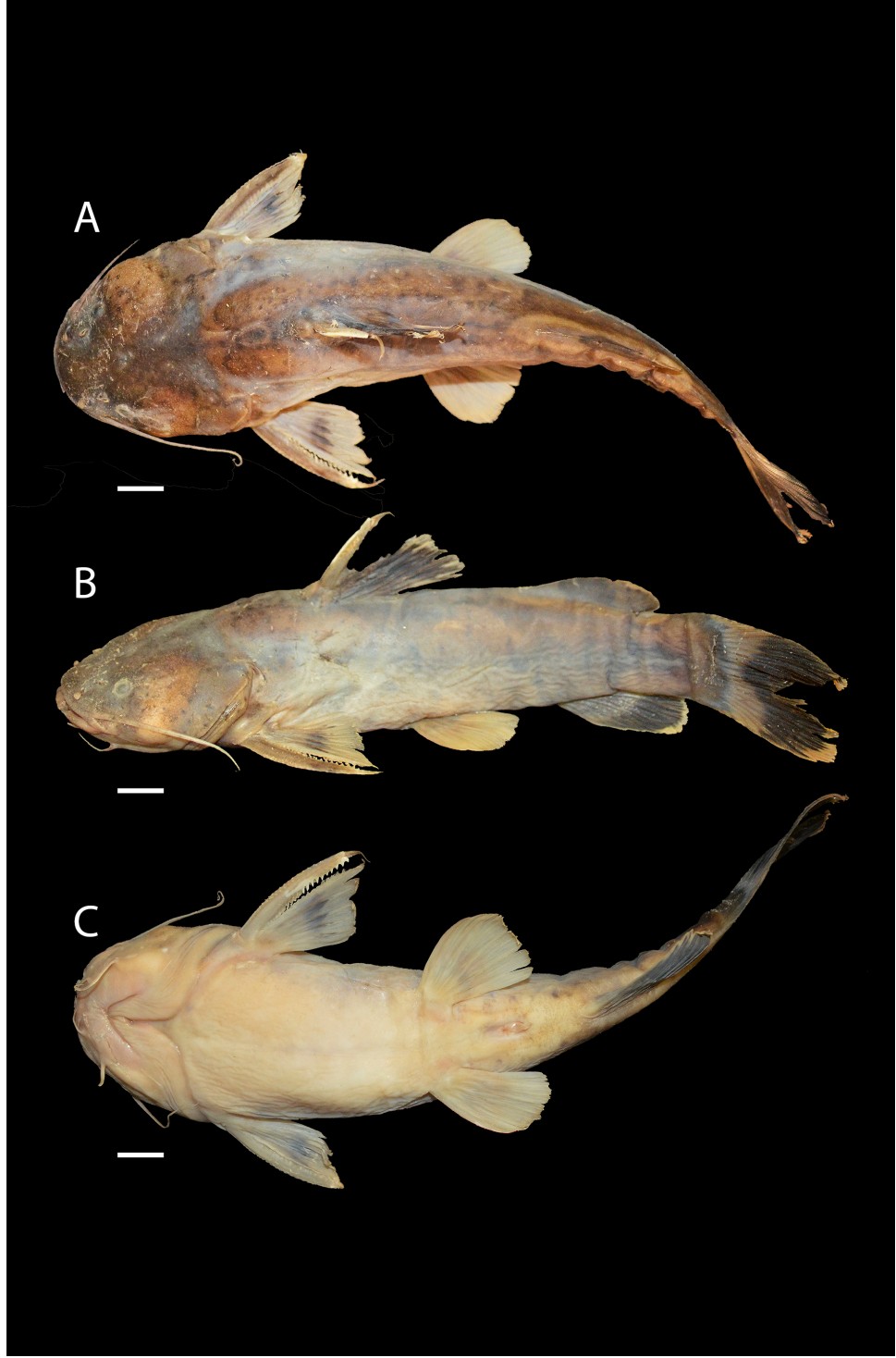

**Figure 9** *Pseudopimelodus atricaudus,* **holotype, CIUA 5141, 143.7 mm SL, Colombia, Sucre, Magdalena River basin, Cauca River in Guaranda.** (A, B, C): Dorsal, lateral, and ventral view, respectively. Scale bar = one cm. Photography: Giovany Olaya Betancur. Photographic edition: Mauricio Urrego Tobón.

Tolima, Magdalena River in Honda, *ca.* 5°14′05″N, 74°43′43″W, 12 Jan 2007, Ortega-Lara A. IMCN 8233, 6, 99.4–275.0 mm SL, Colombia, Córdoba, San Jorge River in La Balsa, Corregimiento La Apartada, *ca.* 8°01′40″N, 75°23′04″W, 20 Jan 2016, Vallecilla-Valencia V., Ortega-Lara A. IMCN 8237, 7, 202–225 mm SL, Colombia, Santander, Magdalena River Basin, Magdalena River in Barrancabermeja, 7°01′56″N, 73°52′35″W, 26 Sep 2011, Ortega-Lara A. IMCN 8238, 1, 62.6 mm SL, Colombia, Bolívar, Magdalena River Basin, Yanacue stream in the Yanacue town, 7°18′14″N, 74°01′16″W, 15 Aug 2010, Ortega-Lara A. IMCN 8266, 1 DS, 203.5 mm SL, Colombia, Antioquia, Magdalena River Basin, Cauca River in Barrio Chino, Caucasia, 8°0′35.3″N, 75°0′0″W, Dec 2015, Restrepo-Gómez A.M. IMCN 8268, 1 DS, 160 mm SL, Colombia, Bolívar, Magdalena River Basin, Cauca River in San Jacinto del Cauca, 8°12′7.5″N, 74°45′15.5″W, Dec 2015, Restrepo-Gómez A.M. IMCN 8271, 1, 198.6 mm SL, Colombia, Bolívar, Magdalena River Basin, Cauca River in Punta Cartagena, Pinillos, 8°53′37.4″N, 74°28′28.5″W, Apr 2015, Olaya G.

**Non-type material**

CIUA 366, 95 mm SL, Colombia, Cesar, Magdalena River Basin, Tucuy River in Becerril, 9°41′35″N, 73°27′42.1″W, May 2006, Montoya A.F. CIUA 367, 120 mm SL, Colombia, Cesar, Magdalena River Basin, Maracas River in Becerril, 9°44′46.2″N, 73°10′38″W, May 2006, Montoya A.F. CIUA 528 3, 210–245 mm SL, Colombia, Valle del Cauca, Cauca River Basin, in Río Frío, 4°07′52.4″N, 76°16′22.1″W, Jan 2007, Ochoa L., Ospina J.G. CIUA 840, 8 DS, Colombia, Antioquia, Magdalena River Basin, Magdalena River in Puerto Berrío, 6°30′27.8″N, 74°23′49.3″W, Aug 2006, Cano J.M. CIUA 841, 7 DS, Colombia, Antioquia, Magdalena River Basin, Magdalena River in Puerto Berrío, 6°30′27.8″N, 74°23′49.2″W, Aug 2006, Cano J.M. CIUA 1151, 4, 135–160 mm SL, Colombia, Antioquia, Magdalena River Basin, Magdalena River in Puerto Berrío, 6°30′27.8″N, 74°23′49.3″W, Aug 2009, Jiménez L.F. CIUA 2029, 2, 81.2–82.1 mm SL, Colombia, Santander, Magdalena River Basin, Sogamoso River in Betulia, 7°05′14.2″N, 73°23′52″W, Jan 2011, Pelayo P. CIUA 2987, 195 mm SL, Colombia, Santander, Magdalena River Basin in El Llanito floodplain lake, 7°10′12.2″N, 73°51′43.8″W, Aug 2010, Carvajal J.D., Hernández A. CIUA 3240, 250, 200 mm SL, Colombia, Antioquia, Magdalena River Basin, Magdalena River in Puerto Berrío, 6°30′27.8″N, 74°23′49.3″W, Jun 2013, Jiménez L.F. CIUA 3291, 2, 145–190 mm SL, Colombia, Antioquia, Magdalena River Basin, Magdalena River in Puerto Berrío, 6°30′27.8″N, 74°23′49.2″W, Jun 2013, Jiménez L.F. CIUA 3668, 190 mm SL, Colombia, Antioquia, Magdalena River Basin, Magdalena River in Puerto Berrío, 6°29′50″N, 74°23′53″W, Aug 2014, Jiménez L.F. CIUA 4797, 190 mm SL, Colombia, Antioquia, Magdalena River Basin, Magdalena River in Puerto Berrío, 6°29′50″N, 74°23′53″W, May 2015, Londoño J.

**Diagnosis.** *Pseudopimelodus atricaudus* (Fig. 9) differs from the other *Pseudopimelodus* species by having a dark band covering $\frac{3}{4}$ of the caudal fin with base of rays and tip of caudal-fin lobes hyaline (*vs.* narrow vertical dark band along the center of caudal fin in *P. bufonius*, *P. magnus*, and *P. schultzi,* although in some specimens is hardly visible; base of rays hyaline and broad band covering $\leq \frac{1}{2}$ caudal fin in *P. charus* and *P. mangurus*), anterior margin of the dorsal-fin spine smooth (*vs.* serrated; Fig. 10). *Pseudopimelodus*

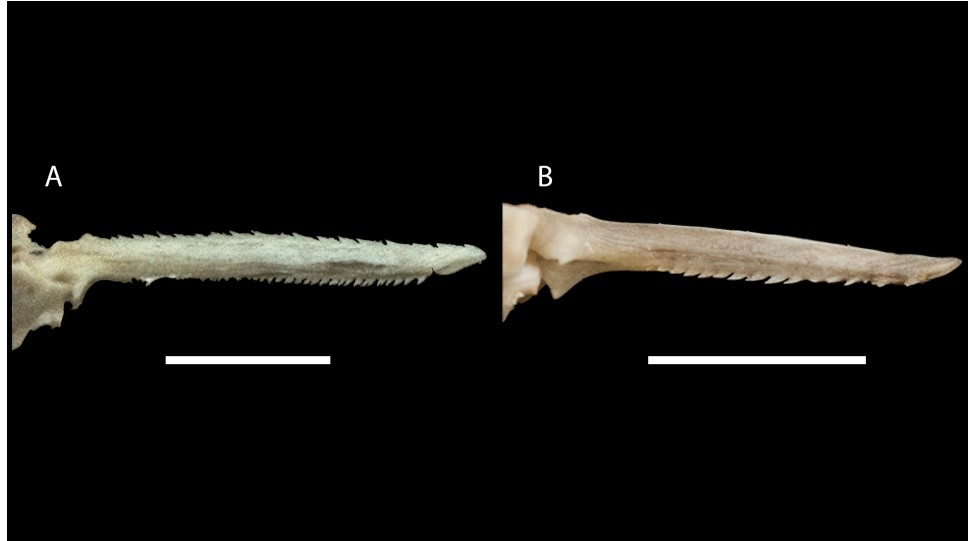

**Figure 10** **Dorsal-fin spine of *Pseudopimelodus magnus*, paratype, IMCN 8265, 213.7 mm SL (A) and *P. atricaudus,* paratype, IMCN 8266, 203.5 mm SL (B).** Scale bar: one cm. Photography: Anderson Cardona Ruiz. Photographic edition: Mauricio Urrego Tobón.

*atricaudus* differs from other *Pseudopimelodus* species except *P. mangurus* by having a total of 39 vertebrae (*vs*. 38 in *P. bufonius*; 43–44 in *P. magnus*, Fig. 3; 41 in *P. schultzi*). It differs from *P. magnus* by having a shallowly concave lateral margin of the transverse process of the fourth vertebra of the Weberian complex (Fig. 4 *vs*. deeply acute notch). Junction of the posterior margin of the transverse process of the fourth vertebra at angle approaching 90°(*vs*. < 90°; Figs. 4 and 5). It differs from *P. magnus*, *P. bufonius*, and *P. mangurus* by having a longer Weberian complex in relation to the length of the neurocranium (46.9–55.5%, Fig. 5 *vs*. 32.1–36.3% in *P. magnus*, 33.5% in *P. bufonius,* and 42.7% in *P. mangurus*). It differs from *P. bufonius* and *P. mangurus* by having shorter maxillary barbels, not surpassing the opercular margin (*vs*. surpassing). It differs from other *Pseudopimelodus* species except *P. magnus* by the length of the posterior process of cleithrum (1.25–1.62 times the width of the pectoral-fin spine base *vs*. 0.57–0.75 in *P. bufonius*, 2.22–2.76 in *P. schultzi*, 2.24–2.30 in *P. mangurus*). It differs from other *Pseudopimelodus* species except *P. magnus* and *P. schultzi* by having a heart-shaped gas bladder (*vs*. dumbbell-shaped; Fig. 6). **Description**. Morphometric data in Table 3. Body depressed from snout to dorsal-fin origin; progressively compressed towards caudal-fin base. Snout rounded in dorsal view. Head as long as wide and covered by thick skin hiding fontanel and cranial roof bones. Eye small, covered by skin and positioned latero-dorsally. Prognathous jaw. Teeth small and villiform. Premaxilla laterally projected backwards, surpassing lateral process of lateral ethmoid. Anterior nostril tubular, located lateroposteriorly to maxillary barbel base (Figs. 8 and 10). Distance from anterior nostril to eye greater than distance to posterior nostril. Maxillary barbel reaching opercular margin. Mental barbel inserted anterior to gular apex. Inner mental barbel surpassing gular apex. Outer mental barbel surpassing branchial opening. Gular fold V-shaped, with rounded apex (Fig. 8). Branchiostegal membrane free

**Table 3  Morphometric data of *Pseudopimelodus atricaudus* (41 specimens).**

|  | Holotype | Max | Min | Mean | SD |
|---|---|---|---|---|---|
| Standard length (mm) | 143.7 | 278.3 | 87.7 | 173 | 38.7 |
| **Percentages of standard length** | | | | | |
| Head length | 30.1 | 33.4 | 26.3 | 29.9 | 1.8 |
| Pre-dorsal distance | 37.5 | 40.6 | 35.2 | 37.6 | 1.5 |
| Pre-pectoral distance | 29.5 | 30.8 | 24.6 | 26.8 | 1.6 |
| Pre-pelvic distance | 52.9 | 59.6 | 51.3 | 54.9 | 2.2 |
| Distance between pectoral-fin origin and dorsal-fin origin | 23.9 | 27.4 | 21.6 | 24.7 | 1.2 |
| Distance between pectoral-fin origin and pelvic-fin origin | 25.6 | 37.7 | 25.6 | 32.3 | 2.5 |
| Distance between dorsal-fin origin and pelvic-fin origin | 25.7 | 30.5 | 21.6 | 26.5 | 2.1 |
| Dorsal-fin base length | 14.8 | 16.9 | 13.1 | 14.8 | 0.8 |
| Distance between adipose-fin origin and pelvic-fin origin | 24.8 | 36.8 | 24.1 | 29.1 | 2.4 |
| Distance between pelvic-fin origin and anal-fin origin | 23.4 | 30.5 | 21.8 | 24.7 | 1.7 |
| Distance between anal-fin origin and adipose-fin origin | 17.9 | 19.0 | 14.4 | 16.9 | 1.2 |
| Adipose-fin base length | 17.8 | 17.8 | 11.9 | 15.1 | 1.6 |
| Anal-fin base length | 11.7 | 13.3 | 9.5 | 10.8 | 0.9 |
| Pectoral-fin base length | 23.7 | 32.1 | 20.4 | 25.5 | 3.1 |
| Pelvic-fin base length | 17.2 | 23.3 | 13.4 | 17.6 | 2.2 |
| **Percentages of head length** | | | | | |
| Orbital diameter | 8.2 | 11.1 | 5.6 | 8.3 | 1.3 |
| Snout length | 36.8 | 40.4 | 20.7 | 30.8 | 4.8 |
| Distance between maxillary barbels | 45.3 | 56.5 | 42.2 | 48.9 | 3.3 |
| Distance between anteriormost mesial point of snout and anterior nostril | 9.5 | 16.0 | 6.5 | 10.1 | 2.5 |
| Distance between maxillary barbel and eye | 11.3 | 18.1 | 10.9 | 14.3 | 1.6 |
| Distance between anterior nostrils | 26.9 | 30.5 | 20.9 | 25.8 | 2.3 |
| Distance between posterior nostrils | 30.2 | 33.5 | 21.3 | 28.6 | 2.7 |
| Distance between anterior and posterior nostrils | 7.1 | 9.4 | 4.2 | 7.0 | 1.0 |
| Distance between posterior nostril to eye | 6.9 | 10.5 | 6.4 | 8.6 | 1.2 |
| Interorbital distance | 43.0 | 49.4 | 34.5 | 43.2 | 3.9 |
| Mouth width | 67.2 | 93.7 | 60.8 | 77.2 | 7.6 |
| Distance between outer mental barbels | 47.8 | 51.2 | 38.6 | 44.4 | 2.9 |
| Distance between inner mental barbels | 25.1 | 31.7 | 18.7 | 25.4 | 2.5 |

from isthmus. Posterior process of cleithrum triangular, 1.25–1.62 of width of pectoral-fin spine base.

Vomer T-shaped, with bifurcated posterior process (Fig. 4B) and in contact with parasphenoid, mesethmoid, and lateral ethmoid. Posterior region of mesethmoid wider than base of parieto-supraoccipital process. Lateral margin of transverse process of the fourth vertebra of Weberian complex smoothly-concave joining posteriorly to vertebral centra at right angle. Length of Weberian complex in relation to length of neurocranium 32.2–36.3%. Anterior fontanel elongated, reaching a transverse line through anterolateral process of lateral ethmoid (Fig. 5B). Posterior fontanel small and oval-shaped, located

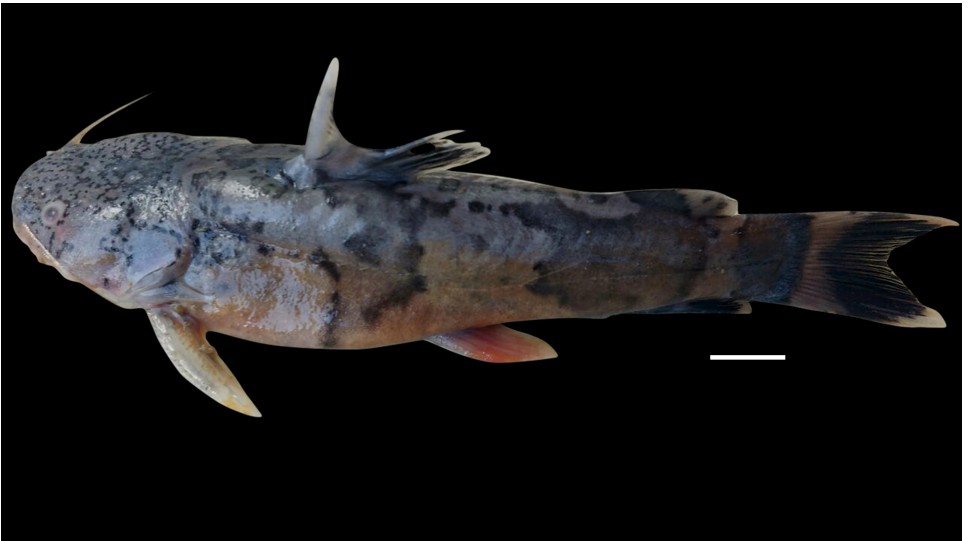

**Figure 11 Juvenile live individual of *Pseudopimelodus atricaudus* (not collected) from Colombia, Sucre, Cauca River in Guaranda.** Scale bar: one cm. Photography: Ana M. Restrepo Gómez. Photographic edition: Mauricio Urrego Tobón.

at center of parieto-supraoccipital (Fig. 5B). Parieto-supraoccipital process rectangular, slightly wider at base, with bifurcated tip in contact with supraneural (Fig. 5B). Heart-shaped gas bladder (Fig. 6). Lateral line complete, reaching caudal-fin base. Total number of vertebrae 39. Dorsal-fin origin at anterior third of body, posterior margin straight, dorsal-fin spine strongly ossified, of equal length to first branched ray. Anterior margin of dorsal-fin spine smooth with its distal end pointed. Dorsal-fin rays I, 6. Adipose-fin origin at level of anal fin origin. Pectoral-fin spine with serrations and covered by skin, anterior and posterior margins with 17–19 and 13–17 serrations, respectively. Serrations on posterior margin slightly larger than those on anterior margin. Posterior edge of pectoral fin straight. Axillary pore present. Pectoral-fin rays I, 7. Pelvic fin emarginated, inserted behind end of dorsal-fin base. Pelvic-fin rays i, 5. Anal fin with rounded posterior edge and inserted below 17th vertebra. Anal-fin rays v, 7. Caudal-fin bifurcated, with pointed lobes, upper lobe slightly narrower and longer than ventral lobe. Caudal-fin rays i, 8, 7, i; dorsal procurrent rays 17; ventral procurrent rays 17.

**Color in life and alcohol.** Body covered by yellowish mucus with four irregularly-shaped dark bands on pre-dorsal, subdorsal, sub-adipose region, and caudal-fin base, clearly visible in juveniles (Fig. 11), but faint in adult specimens. Pre-dorsal and subdorsal bands not joined. Remaining bands variably joined at different levels without defined pattern. Body with small dark and scattered spots and belly whitish. Dorsal and anal fins brown with posterior margin pale. Caudal-fin base hyaline, dark brown on its posterior three-quarters. Pectoral and pelvic fins reddish with dark band at base. Color in alcohol similar to color in life.

**Distribution.** It is found on the Magdalena River below 200 m asl from the confluence of the Guali River in the city of Honda and in the middle and lower basins of the Cauca River, close to the mouth (Fig. 8).

**Etymology.** The specific name *atricaudus* is from the Latin, "ater" (black) and "cauda" (tail) and refers to the diagnostic black caudal fin.

## DISCUSSION

Molecular and morphological approaches were used in this study to test the hypothesis that undescribed species of *Pseudopimelodus* occur in Colombia. Available COI sequences allowed phylogenetic comparisons of samples studied here with members of all *Pseudopimelodus* species and other genera of Pseudopimelodidae. In contrast to the remaining *Pseudopimelodus* species, some osteological features remained unaddressed in *P. charus,* due to the lack of skeletonized specimens, although we could analyze its external morphology and coloration patterns.

This study provides evidence of two new species of *Pseudopimelodus* from the Magdalena-Cauca River Basin, which due to similarities in the external morphology and lack of adequate taxonomic studies, were formerly identified as *P. bufonius* or *P. schultzi* (see synonymy of new species). *Pseudopimelodus magnus* shows a well-supported sister group relationship with *P. schultzi,* whereas *P. atricaudus* is the sister group to all *Pseudopimelodus* species. Additionally, the genetic distances in *Pseudopimelodus* species (see section molecular analyses) are concordant with intraspecific variation reported in Siluriformes (*Hubert et al., 2008*), except for *P. bufonius* from the Orinoco River Basin that showed larger genetic distances from its conspecifics from the Vaupés and Orteguaza rivers. A recent study indicates that *P. atricaudus* (lineage 1) diverged from *P. magnus* (lineage 5) about 16 mya, resulting from the uplift of the Antioqueño Plateau (*Rangel-Medrano, Ortega-Lara & Márquez, 2020*).

The new *Pseudopimelodus* species present three synapomorphies (thick skin on pectoral-fin spine, tip of pectoral-fin spine bifurcated and, small pseudotympanum opening) for the clade [*Cruciglanis*, *Pseudopimelodus*] *Rhyacoglanis*] and two synapomorphies for *Pseudopimelodus* (triangular mesacoracoid and a short posterior cleithral process) previously reported by *Shibatta & Vari (2017)*. Additionally, juveniles of *P. atricaudus* shows clearly visible bands while adult specimens exhibit faint or no bands with small dark and scattered spots, supporting one of the synapomorphies of Pseudopimelodidae.

However, the findings of heart-haped gas bladder in the two new species described herein, *P. mangurus*, and *P. schultzi*, do not provide support to the dumbbell-shaped gas bladder (*Birindelli & Shibatta, 2011*) as synapomorphy of the clade [*Cruciglanis*, *Pseudopimelodus*] *Rhyacoglanis*] (*Shibatta & Vari, 2017*). Compared with *Shibatta & Vari (2017)*, this study also found a wider range of number of vertebrae (38–44 *vs.* 41–42), partial fusion between predorsal and subdorsal bands (*vs.* absent), and short maxillary barbels, falling short of vertical through dorsal-fin origin, although they are shorter in *P. magnus* and do not reach the opercular margin in *P. atricaudus*.

Moreover, the anteriorly smooth dorsal-fin spine in *P. atricaudus* contrasts with the anteriorly serrated dorsal-fin spine described in *Pseudopimelodus* and *Rhyacoglanis* (*Shibatta*

*& Vari, 2017*). This character, along with the pigmentation pattern of the caudal fin, represent autapomorphies for *P. atricaudus*. The largest body size and the highest number of vertebrae allow the discrimination of *P. magnus* from congeners, including *P. mangurus*, the longest *Pseudopimelodus* species known until now (see *Shibatta, 2003*). Likewise, the angles formed, in ventral view, between anterior and lateral margins of transverse process of the fourth vertebra of the Weberian complex also provided a diagnostic trait for *P. magnus*.

*Pseudopimelodus* now includes six species (*P. atricaudus*, *P. bufonius*, *P. charus*, *P. magnus*, *P. mangurus*, *P. schultzi*), although this number is expected to increase, given the high genetic divergence observed among its members (*Rangel-Medrano, Ortega-Lara & Márquez, 2020*). The high genetic divergence observed among specimens from the Sinú River (type locality of *P. schultzi*) and other *Pseudopimelodus*, suggests that *P. schultzi* is restricted to only the Caribbean river basin and therefore its presence in the Magdalena-Cauca River Basin must be questioned. These results contribute to redefine the geographical distribution of *Pseudopimelodus* in northwestern South America and may support further studies in phylogenetics, fisheries, reproduction, and population genetics to eventually elucidate the basic biology of these species and their conservation status.

## CONCLUSIONS

Based on molecular and morphological analyses, this study describes two new species of *Pseudopimelodus* Bleeker, 1858 (Siluriformes: Pseudopimelodidae) from the Magdalena-Cauca River Basin, Colombia. These results show the taxonomical complexity of *Pseudopimelodus* in an area where the presence of only two of four members of this genus have been recorded. Thus, further taxonomic as well as phylogenetic studies of *Pseudopimelodus* are needed to clarify the status of divergent lineages currently included in this genus.

## ACKNOWLEDGEMENTS

The authors thank Fundación para la Investigación y el Desarrollo Sostenible (FUNINDES) for logistical support and facilities, IMCN, INCIVA, for allowing examination of specimens from the reference collection at their facilities. To Darlyn Fabiola Mosquera, Carlos Ardila-Rodríguez, Viki Vallecilla and Libardo Tapiero, for collection of part of the studied material. To Raúl Ríos (IMCN), Luz Fernanda Jiménez (GUIA), Francisco Villa-Navarro (CZUT-P), Iván Mojica (ICN-MHN), Carlos Ardila-Rodríguez (CAR), Tulia Rivas (UTCH-P), and Jonathan Armbruster (AUM), for the loan of specimens and logistical support while visiting collections. Authors also thank Anderson Cardona Ruiz, Giovany Olaya Betancur and Mauricio Urrego Tobón for their assistance with photographs; Donald Taphorn and the anonymous reviewers for their comments, which improved the final version of this article.

### Funding

This work was funded by the Universidad Nacional de Colombia, Sede Medellín and Empresas Públicas de Medellín, Grant CT-2013-002443 ''Variación genotípica y fenotípica de poblaciones de especies reófilas presentes en el área de influencia del proyecto hidroeléctrico Ituango'' and by Grant CT2019000661 ''Variabilidad genética de un banco de peces de los sectores medio y bajo del río Cauca''. The funders had no role in study design, data collection and analysis, decision to publish, or preparation of the manuscript.

### Grant Disclosures

The following grant information was disclosed by the authors:
Universidad Nacional de Colombia.
Sede Medellín and Empresas Públicas de Medellín: CT-2013-002443.
Variación genotípica y fenotípica de poblaciones de especies reófilas presentes en el área de influencia del proyecto hidroeléctrico Ituango: CT2019000661.
Vicerrectoría de Investigaciones y Programa de Doctorado en Ciencias-Biología.
Facultad de Ciencias, Universidad del Valle, Cali, Colombia: Convocatoria Interna 2017, CI 71099.

### Competing Interests

The authors declare there are no competing interests.

### Author Contributions

- Ana M. Restrepo-Gómez conceived and designed the experiments, performed the experiments, analyzed the data, prepared figures and/or tables, authored or reviewed drafts of the paper, performed the morphological analyses, and approved the final draft.
- José D. Rangel-Medrano conceived and designed the experiments, performed the experiments, analyzed the data, prepared figures and/or tables, authored or reviewed drafts of the paper, performed the molecular analyses, and approved the final draft.
- Edna J. Márquez conceived and designed the experiments, analyzed the data, prepared figures and/or tables, authored or reviewed drafts of the paper, molecular analysis, and approved the final draft.
- Armando Ortega-Lara conceived and designed the experiments, performed the experiments, analyzed the data, prepared figures and/or tables, authored or reviewed drafts of the paper, performed the morphological analyses, and approved the final draft.

### Data Availability

COI sequences are available at GenBank: COI sequences are available at GenBank: MH553571, MH800619–MH800632, MH800634–MH800639, MH553581, MH553580, MH553583, MH553584, MH553586, MH800809–MH800812, MH800711–MH800715, MH553589, MH553590, MH553593, MH553594, MH553595, MH553596, MH553597, MH553598, MH553599, MH800832, MH553609, MH553607, MH553605–MH553606, MH553604, MH553610, MH553611.

## New Species Registration

The following information was supplied regarding the registration of a newly described species:

Publication LSID: urn:lsid:zoobank.org:pub:8B78D766-07A3-47A8-A12C-F55958703ACB

*Pseudopimelodus magnus* sp. nov. LSID: urn:lsid:zoobank.org:act:569031C8-163A-4E61-8539-44CDD3F09592

*Pseudopimelodus atricaudus* sp. nov. LSID urn:lsid:zoobank.org:act:6BF0B744-2D8D-4AC5-A9E3-69D1213A1965.

## Supplemental Information

Supplemental information for this article can be found online at http://dx.doi.org/10.7717/peerj.9723#supplemental-information.

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
