# Peer review of "Two new species of Pseudopimelodus Bleeker, 1858 (Siluriformes: Pseudopimelodidae) from the Magdalena Basin, Colombia"

_PeerJ, doi:10.7717/peerj.9723_

## Round 0.1 · original submission · Minor Revisions

Dear Dr. Restrepo-Gómez and colleagues:

Thanks for submitting your manuscript to PeerJ. I have now received two independent reviews of your work, and as you will see, both are favorable. Well done! Nonetheless, both reviewers raised some minor concerns about the research, and areas where the manuscript can be improved. Be aware that reviewer 1 provided a marked-up copy of your manuscript in addition to general criticisms. Please address reviewer 2’s concern regarding the illustrations of the specimens (live and alcohol-preserved) and dry skeletons.

I agree with the concerns of the reviewers, and thus feel that their criticisms should be adequately addressed before moving forward.

Therefore, I am recommending that you revise your manuscript accordingly, taking into account all of the issues raised by the reviewers. I do believe that your manuscript will be ready for publication once these issues are addressed.

Good luck with your revision,

-joe

Reviewer 1 ·

Basic reporting

The article was clearly written but still have some minor grammar issues, pointed in the included commented version.
Sufficient and comprehensive use of references.
Comprehensible introduction with a nice review of the taxonomic situation of the group.
Good quality figures with some minor details on the labeling. Recommendations included in the text.

Experimental design

Research question well defined, relevant & meaningful.
The description, as well as the data supporting new species, is rigorous.
Methods were described in detail, although it can be improved, especially the section related to the substitution models and genetic distance.

Validity of the findings

Findings are novel and they have profound implications on the taxonomy of this fishes in the region.
Data is robust, however, there are some statements on the discussion that needs to be clarified still the conclusions are not affected.

Additional comments

This paper, with the correction of some details, will undoubtedly become a nice contribution to the fish fauna of the region. It is well presented, clearly written and data is robust. It still needs some improvement on the methods, specifically on the phylogenetic analysis section regarding the use of substitution models and genetic distance correction.

Annotated reviews are not available for download in order to protect the identity of reviewers who chose to remain anonymous.

Reviewer 2 ·

Basic reporting

see below

Experimental design

SEE BELOW

Validity of the findings

see below

Additional comments

This is a very interesting paper that describes two new species of Pseudopimelodus. The species hypotheses are strong as they are based on morphological and molecular characters. However, the illustrations of the specimens (live and alcohol-preserved) and dry skeletons are very poor and should be replaced. In addition, the paper uses a restricted sample of Colombian specimens/species that make the discussion and Diagnosis limited, what should be noted and properly included in the text. Therefore, I recommend this paper to be accepted for publication after corrections are made (see below for corrections). Once these modifications are incorporated, I see no reason for additional outside review.

Minor corrections:
Introduction: lines 61 to 69. Authors cite the synapomorphies of Pseudopimelodidae, citing Shibatta & Vari (2017), and then in the next paragraph, cite Diogo et al. (2014) saying ´in a further analysis´. Times or citations are not correct.
Introduction: lines 86, 87. Authors did not cite the study that formally report two species of Pseudopimelodus in the Magdalena.
Introduction, lines 98, 99. osteological should be considered under morphological analyses. Perhaps authors should say: molecular and morphological analyses, including osteology and external morphology.
Introduction, line 105: The name of the new species should be kept to the RESULTS.
Results. lines 187 to 218. Paratypes should be better distributed. Samples should be sent to Brazilian institutions (at least).
Results. Line 221. Authors should provide N for number of vertebrae in P. mangurus and P. charus. Shibatta & Vari do not provide the precise count of vertebrae in those two species.
Results. Line 224. Authors did not examine skeletons of all species of Pseudopimelodus (e.g., P. charus), and therefore can not say the shape of the transverse process of the fourth vertebra in all species of Pseudopimelodus.
Results. Line 228. Use gas bladder description in Birindelli & Shibatta (2013), instead of lobed.
Results. Descriptions are too short. See published descriptions (e.g., Shibatta & Vari, 2017).
Discussion. Lines 387, 388. Authors did not include all species of Pseudopimelodus in their phylogenetic analysis (e.g., P. charus), and therefore should rephrase some sentences accordingly.
Results. Lines 267, 268. Authors should note that Pseudopimelodus mangurus reaches up to at least 345 mm SL (Shibatta, 2003), and therefore is larger than P. magnus.
Table 1. Samples of Pseudopimelodus mangurus are missing from the table, even though present in the analysis.
List of Comparative Material should be provided.

Reviewer 3 ·

Basic reporting

Experimental design

Validity of the findings

Additional comments

All my suggestions, corrections, and comments are directly indicated into the pdf file (attached).

The manuscript basically presents compelling evidence from barcoding and morphological analyses that support recognition of two new species of Pseudopimelodus from the trans-Andean basins of Colombia.

However, there are several problematic issues regarding the following:

Inconsistency along the text to keep a standardized format in the lists of examined lots of specimens, lists of types, and geographic coordinates.

English is not fluent and construction of some paragraphs is awkward, resulting in statements or ideas difficult to understand, even for the native English speaker. Also, there are some Spanish words in the text.

Authors cite some results from unpublished research that is not accessible to the reader and some conclusions advanced in the manuscript are entirely based on this unpublished evidence.

Some important and pertinent references (i.e. Shibatta & Vari, 2017; Abrahão et al., 2018) on the taxonomic group, subject of this manuscript (Pseudopimelodidae) are not cited. These references will provide a crucial framework for the discussion of the presented results. I guess these references were inadvertently omitted by the authors. Therefore, I ask them to carefully review those works, in order to present a current and complete discussion of their results.

Regarding the selection of a couple of institutions (ANDES, CAR) as repositories of type specimens, there are some environmental law restrictions in Colombia, prohibiting that type specimens are deposited in institutions that are not registered in the National Registry of Biological Collections (http://rnc.humboldt.org.co/wp/).

Some illustrations or photographs are required to show some proposed diagnostic character-states of the newly described species.

Taxonomic descriptions are extremely brief and general and do not provide enough anatomical data, according to current standards in the taxonomy of the group (Pseudopimelodidae).

Geographic distribution sections are not informative enough on the actual geographic distribution of the species being treated.

Complete lists of synonyms (cresonymy) must be provided for each species name.

Some technical problems with the figures are detailed in the manuscript and have to be solved.

Some abbreviations are missing. They have to be explained in the Materials and Methods section.

Figure 7 must be corrected or replaced by a figure showing the actual geographic distribution of the species described in the paper.

Misleading interpretation of some results are indicated in the text of the file.

As result of my review this manuscript needs an exhaustive process of revision by the authors and needs to be submitted again for a second round of review.

Hoping my suggestions are helpful for improvement of the scientific quality of the manuscript.

Annotated reviews are not available for download in order to protect the identity of reviewers who chose to remain anonymous.

---

## Round 0.2 · Minor Revisions

Dear Dr. Restrepo-Gómez and colleagues:

Thanks for resubmitting your manuscript to PeerJ. I have now received two independent reviews of your work (from the original reviewers), and as you will see, both are still favorable and agree that the resubmission is much improved. Well done! However, both reviewers raised a few more minor concerns about the research, and areas where the manuscript can still be improved. NOTE: both reviewers provided a marked-up copy of your manuscript in addition to general criticisms. Please address all of the new concerns raised by both reviewers.

I agree with the concerns of the reviewers, and thus feel that their criticisms should be adequately addressed before moving forward.

Therefore, I am recommending that you revise your manuscript accordingly, taking into account all of the issues raised by the reviewers. I do believe that your manuscript will be ready for publication once these issues are addressed.

Good luck with your revision,

-joe

Reviewer 1 ·

Basic reporting

I think authors have made a good job in considering suggested comments and corrections. English was indeed improved as well as references and many other proposed changes.

Experimental design

I just have to criticize the use of data substitution models taken from other datasets and not from your own. This may not change results substantially but I consider this to be a flawed use of standard methods.

Validity of the findings

Findings are well supported on varied sources of evidence.

Additional comments

I have still some very minor comments tracked and indicated in the revised version of the manuscript included.

Annotated reviews are not available for download in order to protect the identity of reviewers who chose to remain anonymous.

Reviewer 3 ·

Basic reporting

The use of the English language is clear overall. Still, there are some instances of unusual or awkward sentences that should be fixed.

It is reiterative the use of citations to unpublished research results in this version of the manuscript that should be avoided or deleted at all. Only results obtained from the present study should be described and discussed.

Some ambiguity arise in the terminology applied for osteological structures from the use of different bibliographic references that do not follow a unique nomenclature (Lundberg and McDade, 1986 vs. Arratia, 2003a, b).

Some statements of the Abstract, Introduction, and Discussion sections are misleading, and they are individually commented on the manuscript file.

List of comparative material have to be fixed.

Bibliographic reference of the illustration of Pseudopimelodus charus as source for comparisons with this species has to be corrected.

The Weberian complex length is not explained.

Synonymy lists have to be carefully revised. Authors do not follow a unique format. They mixed up bibliographic references with lots and several entries lack of the respective subject (e.g. Identification key, distribution, fisheries).
I recommend to check the following link (http://www.scielo.br/revistas/ni/iinstruc.htm), where a detailed explanation on how to assemble a synonymy section for a species account is provided.
Also, the authors can consult the following reference for a good example of synonym format:
Vari, R. P., C. J. Ferraris, Jr. and M. C. C. de Pinna. 2005. The neotropical whale catfishes (Siluriformes: Cetopsidae: Cetopsinae), a revisionary study. Neotropical Ichthyology 3 (2): 127-238.

Figure legends are not provided in the manuscript and some figures required addition of arrows to better indicate specific structures used as diagnostic characters (e.g. acute notch on the lateral margin of the anterior arm of the fourth transverse process of the Weberian complex).

The relative extension of the dark band of the caudal fin, used as diagnostic character is ambiguous for some species (e.g. Pseudopimelodus charus).

Osteological characters used in the diagnosis have to be incorporated into the description.

A wealth of morphological characters used in the context of recently published phylogenetic analyses are worthy to be incorporated into the study, especially considering that some emphasis is given to the placement of the new species at the genus level. Some morphological synapomorphies have been proposed for Pseudopimelodus (Shibatta and Vari, 2017) and consequently, their verification in the new species is highly relevant for this study.

Descriptions have to be revised in order to accept all corrections and to incorporate suggestions.

Geographic distribution of each species needs a major revision, since many political entities are included that are not familiar to foreign researchers. Geographic references as rivers (sub-basins or drainage systems) should be used instead of political divisions as departments or municipalities.

Instead of describing morphological structures uninformative for the taxonomy or phylogenetic relationships of Pseudopimelodus species, the authors should include morphological description of the several characters informative on the phylogenetic relationships at different levels of Pseudopimelodus species (see description of character used in the phylogenetic analysis of Shibatta and Vari, 2017).

An appropriate discussion (i.e. comparison of results from different studies) is not reached, since relevant papers on the subject are unconnected with the results obtained in the present paper. An emphasis must be given to try to compare relevant results of independent studies. For example, authors insist that their new species should be described within Pseudopimelodus, but all characters to define this genus were not evaluated in the new species proposed.

Experimental design

The authors cite an unpublished research to justify recognition of two undescribed species, basically as result of a multilocus analysis that is not described in the present manuscript.

Validity of the findings

No comment.

Additional comments

Extensive corrections, comments and suggestions are directly included in the attached files of the manuscript.

A figure legend have to be included to the manuscript.

Labeling of structures in figures 4 and 5 is scanty.

Figure 6 is taxonomically uninformative.

Annotated reviews are not available for download in order to protect the identity of reviewers who chose to remain anonymous.

---

## Round 0.3 · Minor Revisions

Dear Dr. Restrepo-Gómez and colleagues:

Thanks for revising your manuscript. The reviewers are very satisfied with your revision (as am I). Great! However, there are a few minor edits to make. Please address these ASAP so we may move towards acceptance of your work.

Best,

-joe

Reviewer 1 ·

Basic reporting

The use of the language is correct for me, considering that I'm not an native English speaker. I made few corrections to some parts added in this new version.

Experimental design

Well defined questions. Good use of molecular and morphological traits to describe the species. Very through morphological description of both species.

Validity of the findings

Both species are valid and the impact on the native knowledge of the freshwater fishes of the country is going to be positively impacted
Data is clear and available in tables for morphological measurements and sequences for molecular data.

Additional comments

The paper still have some minor details that need your correction mostly in the redaction and language edition in the parts that were added or corrected. They are indicated on the included commented version of the ms.

Annotated reviews are not available for download in order to protect the identity of reviewers who chose to remain anonymous.

Reviewer 3 ·

Basic reporting

Grammatical corrections were directly made on the manuscript.
Most pertinent literature references are cited, but a careful revision of citations of works with more than two authors (et al.) have to be made. Also, format of the References section has to be check in detail, following the journal instructions.
et al., vs., and ca. have to be italicized along the text.
A brief definition of lineages 1 and 5 needs to be incorporated into the Materials and methods section, since an unpublished work by Rangel-Medrano et al. is cited instead, being currently unavailable as providing this information.
Yo have to explain how do you measured the length of the Weberian complex (anatomical landmarks used).
The synonymy sections contain a series of references to vague comments that need to be detail. Do these comments refer to taxonomy, geographic distribution, new distribution records, identification keys, complementary descriptions, ecological data?.
I strongly recommend to designate paratypes in better established and more recognized icthyological collections of Colombia: CZUT-IC, IAvH-P, ICN-MHN, MPUJ.
Fate of type material in some of the selected collections is uncertain in the long term, since some of them lack of proper installations, dedicated personnel in charge, and have experienced erratic administration.
It is necessary to standardize the use of dash or hyphens for numerical ranges.
I recommend to include a comparative figure illustrating the shape of the gas bladder of the two new species, since you found that differ from that previously described for the species of Pseudopimelodus.
It is necessary to built a stronger discussion by comparing the general topology obtained in previous hypotheses with your results.
Discussion of random characters uninformative on the monophyly or phylogenetic relationships of Pseudopimelodus have to be replaced by a proper discussion of relevant synapomorphies of clades that contain the genus, as well as autapomorphies proposed to diagnose Pseudopimelodus.
Remove unnecessary Conclusion section.
Change orientation of some figures to adjust to traditional standard in anatomical works.

Experimental design

No comment.

Validity of the findings

No comment.

Additional comments

The manuscript was noticeably improved but still requires some adjustment and relatively easy corrections before acceptance for publication.

Annotated reviews are not available for download in order to protect the identity of reviewers who chose to remain anonymous.

---

## Round 0.4 · accepted · Accept

Dear Dr. Restrepo-Gómez and colleagues:

Thanks for revising your manuscript based on the concerns raised by the reviewers. I now believe that your manuscript is suitable for publication. Congratulations! I look forward to seeing this work in print, and I anticipate it being an important resource for groups studying pseudopimelodid systematics and evolution. Thanks again for choosing PeerJ to publish such important work.

Best,

-joe